# Observation of Enhanced Dissociative Photochemistry in the Non-Native Nucleobase 2-Thiouracil

**DOI:** 10.3390/molecules25143157

**Published:** 2020-07-10

**Authors:** Kelechi O. Uleanya, Rosaria Cercola, Maria Nikolova, Edward Matthews, Natalie G. K. Wong, Caroline E. H. Dessent

**Affiliations:** Department of Chemistry, University of York, Heslington, York YO10 5DD, UK; oku500@york.ac.uk (K.O.U.); rc1274@york.ac.uk (R.C.); bsmni@leeds.ac.uk (M.N.); edmatthews92@googlemail.com (E.M.); ngkw500@york.ac.uk (N.G.K.W.)

**Keywords:** 2-thiouracil, free-radicals, non-native nucleobase, excited states, phototherapy

## Abstract

We present the first study to measure the dissociative photochemistry of 2-thiouracil (2-TU), an important nucleobase analogue with applications in molecular biology and pharmacology. Laser photodissociation spectroscopy is applied to the deprotonated and protonated forms of 2-TU, which are produced in the gas-phase using electrospray ionization mass spectrometry. Our results show that the deprotonated form of 2-thiouracil ([2-TU-H]^−^) decays predominantly by electron ejection and hence concomitant production of the [2-TU-H]· free-radical species, following photoexcitation across the UVA-UVC region. Thiocyanate (SCN^−^) and a *m*/*z* 93 fragment ion are also observed as photodecay products of [2-TU-H]^−^ but at very low intensities. Photoexcitation of protonated 2-thiouracil ([2-TU·H]^+^) across the same UVA-UVC spectral region produces the *m*/*z* 96 cationic fragment as the major photofragment. This ion corresponds to ejection of an HS· radical from the precursor ion and is determined to be a product of direct excited state decay. Fragment ions associated with decay of the hot ground state (i.e., the ions we would expect to observe if 2-thiouracil was behaving like UV-dissipating uracil) are observed as much more minor products. This behaviour is consistent with enhanced intersystem crossing to triplet excited states compared to internal conversion back to the ground state. These are the first experiments to probe the effect of protonation/deprotonation on thionucleobase photochemistry, and hence explore the effect of pH at a molecular level on their photophysical properties.

## 1. Introduction

The canonical nucleobases of DNA and RNA are renowned for their ability to dissipate harmful UV, primarily via ultrafast excited-state relaxation to the ground state, either directly or indirectly through excited singlet states [1,2,3,4,5,6]. Thiobases represent a class of structurally modified nucleobases where an oxygen atom of the carbonyl group is substituted by a sulphur atom. These molecules display dramatically different relaxation dynamics compared to their canonical nucleobase analogues [7,8,9,10,11,12]. Molecules such as 4-thiothymine and 2-thioguanine possess excited states that evolve by intersystem crossing (ISC) on sub picosecond timescales resulting in nearly unity triplet yields [7,8,9,10,11]. The ensuing triplet state constitutes a highly reactive molecule, which by itself or by generating singlet oxygen [9,11], can damage biomolecules within the cell [13,14]. Hence these molecules are of considerable current interest in phototherapeutic applications [9,15,16,17].

Given the growing application of thionucleobases, there has been considerable interest in characterizing their fundamental photophysics. Time-resolved experiments have primarily been conducted in solution using transient absorption spectroscopy [8,9,10,11,18,19,20,21,22,23,24,25]. Gas-phase studies using time-resolved photoionization techniques have provided complementary insight into the intrinsic decay dynamics [19,20], as well as a straightforward comparison against theoretically derived potential energy surfaces [19,26,27,28,29,30,31].

In this work, we explore the intrinsic (i.e., gas-phase) photochemistry of 2-thiolated uracil (2-TU) (Scheme 1). In particular, we aim to characterise the effect of protonation and deprotonation on the excited states and photoproducts of 2-TU by studying the isolated deprotonated ([2-TU-H]^−^) and protonated ([2-TU·H]^+^) ions via laser-interfaced mass spectrometry (LIMS) [32,33,34,35]. These are the first experiments to directly measure the dissociative photochemistry of a thionucleobase. They are also the first to probe the effect of protonation/deprotonation on a thionucleobase photochemistry, and hence explore the effect of pH at a molecular level on 2-TU photophysical properties. The experiments are important in the context of the use of 2-TU both as a photodynamic therapy agent and biochemical labelling agent since local biochemical environments can display variable pH.

The electronic properties of neutral 2-thiouracil (2-TU) have been the subject of a number of recent theoretical and experimental studies [20,25,30,36,37,38]. Crespo-Hernández and co-workers have applied femtosecond broadband transient absorption spectroscopy in aqueous and acetonitrile solutions, while Ullrich and co-workers used time resolved photoelectron spectroscopy to characterise the gaseous excited state dynamics. Complementary theoretical calculations have been conducted by Gonzalez and co-workers [29,30]. The consensus to emerge from this work is that the main excited-state relaxation pathway following initial excitation to the S_2_ state is S_2_ → S_1_ → T_3_ → T_1_, with S_2_ →S_1_ →T_1_ occurring as a minor pathway [30]. Very recently, experimental and computational evidence has been published providing evidence for the existence of two minima within the T_1_ state [25]. Work has also been conducted on the negative ions of 2-TU [39,40]. These studies are motivated by the fact that ionizing radiation can initiate DNA strand breaks via the formation of transient negative ions [41,42].

## 2. Results

### 2.1. Geometric Structures and Time-Dependent Density Functional Theory (TDDFT) Calculations [2-TU-H]^−^ and [2-TU·H]+

Scheme 2 displays the lowest-energy tautomers of [2-TU-H]^−^ calculated at B3LYP/6-311++G(2d,2p) level. Relative energies for these tautomers are displayed in Table 1. Rotational isomers are grouped together, using small case alphabetical labels (e.g., D3a and D3b). The D1 tautomer corresponding to removal of the N1 proton lies substantially lower in energy than the higher-energy tautomers. It will therefore be the only tautomer produced following electrospray in methanol solvent [43]. In solution, this pattern is repeated, so that the D1 tautomer is again predicted to dominate. Vertical detachment energies (VDEs) were calculated for the [2-TU-H]^−^ isomers, with the value for the D1 isomer predicted to be 3.82 eV. This value can be compared to the experimental value for deprotonated uracil of 2.5 eV [44].

Scheme 3 displays the five lowest-energy calculated tautomers of [2-TU·H]^+^, which are in good agreement with the previous results of Nei et al. [45]. Rotational isomers are again grouped together, using small case alphabetical labels (i.e., P1a–P1d). The lowest-energy gaseous tautomers, P1a and P1b, correspond to a pair of *cis* and *trans* enol-enol rotamers, with other tautomers lying at significantly higher energy. The lowest-energy enol-keto tautomer, P2, is predicted to lie 20.25 kJ mol^−1^ higher in energy than P1a. We therefore expect that P1a will dominate the experimental ion ensemble following electrospray, with some P1b also being present.

TDDFT calculations were conducted to aid the assignment of the excited-state spectra presented below, with the calculated excitation spectra for the lowest-energy tautomers of [2-TU-H]^−^ (D1) and [2-TU·H]^+^ (P1a) displayed in Figure 1. The calculated TDDFT excitation spectra for all tautomers, transitions energies, and assignments of the bright transitions (≥0.005 oscillator strength) are included in the Appendix A. For the sets of rotational isomers (e.g., P1a-P1d of [2-TU·H]^+^, and D3a and D3b of [2-TU-H]^−^), the TDDFT spectra are very similar, and would be indistinguishable at our experimental resolution.

The TDDFT calculations predict that the protonated and deprotonated forms of 2-TU will display dramatically different absorption profiles, with [2-TU-H]^−^ absorbing strongly through the UVA, with the primary absorption of [2-TU·H]^+^ occurring at significantly higher energies.

We note that the TDDFT calculations may not accurately predict the spectra of the [2-TU-H]^−^ negative ion well. Dipole-bound excited states, which are common in negative ion systems, are challenging in general for ab initio calculations, and their accurate calculation requires the use of diffuse functionals centred on the dipole-bound orbital [46,47,48]. In addition, any electronic excitations that appear above the electron detachment threshold of an anionic species will correspond to resonance states [49,50]. The accurate theoretical prediction of such states is beyond the scope of the current work, however the TDDFT calculations presented here have been shown to provide a useful guide in interpreting similar experimental results [51,52,53].

### 2.2. Deprotonated 2-Thiouracil

#### 2.2.1. Gas-Phase Absorption Spectrum of Deprotonated 2-Thiouracil

Figure 2 displays the gas-phase absorption (photodepletion) spectrum of [2-TU-H]^−^ measured over the range 3.2–5.3 eV. The spectrum has an absorption onset at 3.2 eV with continuous absorption through to 5.2 eV. A high-intensity absorption band, labelled (I), is evident between 3.2 to 4.2 eV, peaking at λ_max_ = 3.7 eV. This band lies just below the calculated VDE of the lowest-energy D1 tautomer (3.82 eV). From the relative energies presented in Table 1, only the D1 tautomer is expected to be present following electrospray in methanol. This feature is followed by a lower-intensity, broad absorption (II) from 4.2 to 5.2 eV. The overall profile of the [2-TU-H]^−^ gas-phase absorption spectrum is similar to that of other negatively charged molecules and clusters we studied previously. It can be primarily described as a near-threshold dipole-bound excited state (band I) followed by a higher-energy region where electron-detachment dominates (band II) [43]. However, this picture can be complicated by the presence of electronic transitions of the chromophore, which may be evident superimposed on top of these electron detachment features [33,54].

The calculated TDDFT spectrum of [2-TU-H]^−^ (Figure 1a) predicts that [2-TU-H]^−^ has two main electronic transitions which peak at 3.8 and 4.8 eV. However, these absorptions are not evident in our gaseous experimental spectrum presented in Figure 2, probably due to the dominance of electron detachment (Appendix A). In previous anionic systems we studied, these excited states have been more clearly visible in the photofragment production spectra [44,55,56]. We therefore turn to inspecting the photofragmentation channels of [2-TU-H]^−^ to further characterise the excited states and photochemistry.

#### 2.2.2. Photofragmentation of Deprotonated 2-Thiouracil

Figure 3 displays the photofragment difference (laser on–laser off) mass spectrum of [2-TU-H]^−^ photoexcited at 3.6 eV, close to the band I maximum.

[2-TU-H]^−^ produces two photofragments, *m*/*z* 58 and 93, with the *m*/*z* 58 photofragment being more intense. These ionic photofragments have low intensities compared to the parent ion depletion. Both are produced weakly across band I of the photodepletion spectrum, and with negligible intensity across the band II region. This indicates that across the UV, [2-TU-H]^−^ decays predominantly by electron detachment with associated production of the [2-TU-H]· radical (Equation (1)). The *m*/*z* = 58 photofragment (Equation (2)) is assigned to thiocyanate (SCN^−^). We note that this ion has been observed in low-energy dissociative electron attachment to 2-TU, as well as in collisional activated decomposition of 2-thiouridine [39]. While SCN^−^ was observed in higher-energy collisional dissociation (HCD) of [2-TU-H]^−^, the *m*/*z* = 93 ion was not (Appendix A). This indicates that the *m*/*z* 93 ion (Equation (3)) is a solely photochemical product.
[2-TU-H]^−^ + *hv* → [2-TU-H]· + e^−^(1)
→SCN^−^ + C_3_H_3_NO(2)
→*m*/*z* 93 + H_2_S(3)

Table 2 lists the photofragments and corresponding neutral fragments of [2-TU-H]^−^. We note that there are two possible structures of C_3_H_3_NO (Appendix A). One of these is acrylamide, a potent neurotoxin [57], while the other is a potentially harmful free radical species. Further computational results on the observed fragments are presented in the Appendix A.

The photofragment production spectra are displayed in Figure 4b,c and are presented with the gas-phase absorption spectrum for comparison. While SCN^−^ is produced through band I, the full-width half maximum for this feature is 0.31 eV, which is narrower than that of the band I feature (0.43 eV). Indeed, comparison of the spectra presented in Figure 4 reveals that SCN^−^ is produced only through the lower energy region of band I. It is interesting to note that the SCN^−^ production spectrum appears to be broadened on the high-energy side of the peak, possibly due to unresolved vibrational features. This is reminiscent of near-threshold excitation of I^−^·CH_3_I, where the I^−^ photofragment production spectrum contained a vibrational progression in the C-I stretch, originating in the intermediate transient negative ion [58].

The production profile of the *m*/*z* 93 fragment is very similar to that of the *m*/*z* 58 fragment, although it is produced at ~10× lower intensity. Some non-zero production of *m*/*z* 93 is visible in the region between 4.6–5.2 eV, the area where the second bright transition of [2-TU-H]^−^ is predicted to occur (Section 2.2.1). Indeed, there is also very low-level production of *m/z* 58 in this region. The nature of the excited states and photofragmentation pathways of [2-TU-H]^−^ will be discussed further in Section 3.1.

### 2.3. Protonated 2-Thiouracil [2-TU·H]^+^

#### 2.3.1. Gas-Phase and Solution-Phase Absorption Spectra of [2-TU·H]^+^

Figure 5 displays the gas-phase absorption (photodepletion) spectrum of [2-TU·H]^+^ across the UV region. The spectrum displays two resolved bands, which are labelled I and II, with λ_max_ at 4.68 and 5.3 eV, respectively.

It is instructive to compare the gas-phase absorption spectrum of [2-TU·H]^+^ with that of protonated uracil, which has been studied at low resolution by Pedersen et al. [59] and at high resolution by Berdakin et al. [60] In both studies, protonated uracil displayed a spectrum that consisted of two bands, a weaker intensity band between 260 and 317 nm and a stronger intensity band at higher energies from 227–256 nm. These bands were assigned to the presence of two isomers, an enol-keto tautomer for the weaker band and an enol-enol tautomer for the stronger band. This spectral pattern is remarkably similar to the spectral profile of [2-TU·H]^+^ observed here, allowing us to assign band I to the P2 tautomer, and band II to P1 tautomers. Although our calculations did not predict that the P2 tautomer would be present in the gas-phase, it is known that relative tautomer energies for this type of system can be unreliable. Indeed, there is direct evidence from the IRMPD study of Nei et al. on [2-TU·H]^+^ that an enol-keto tautomer such as P2 is present in the gas-phase following electrospray ionisation [45]. It is also important to note that in previous studies where electrospray has been used to transfer similar molecular ions (e.g., protonated nicotinamide) from the solution into the gas-phase, higher-energy tautomers have been observed, possibly due to the kinetics of the electrospray process [35]. Similar effects may therefore occur for the 2-TU system.

We next turn to inspecting the photofragment action spectra to further probe the nature of the two bands evident in gas-phase absorption spectrum of [2-TU·H]^+^.

#### 2.3.2. Photofragmentation of Protonated 2-Thiouracil

Photofragment mass spectra of [2-TU·H]^+^ were obtained at 4.6 and 5.2 eV close to the λ_max_ of bands I and II (Appendix A). Table 3 provides a list of the most intense photofragments, along with proposed structures and accompanying neutral fragments. The *m*/*z* 96 photofragment, which corresponds to loss of an SH radical, is the most intense photofragment in both bands I and II. Other photofragments observed with significant intensities in both bands are *m*/*z* 128 (H atom loss), *m*/*z* 112 (NH_3_ loss), and *m*/*z* 70 (HNCS loss). The *m*/*z* 68 and *m*/*z* 79 photofragments were not observed in the band I region (Appendix A) but were seen in the higher-energy band II region (Appendix A), although with very low intensities. 

HCD (Higher-Energy Collisional Dissociation) was performed on [2-TU·H]^+^ to explore the thermal fragmentation pathways of the electronic ground state, with the results compiled in Table 3. HCD fragmentation of [2-TU·H]^+^ produces the *m*/*z* 112 (NH_3_ loss) and *m*/*z* 70 (HSCN loss) ions as the dominant products at moderate collision energies which should correspond to internal ion energies close to the photon energies employed in this study [54]. These product ions are in line with those observed for CID of protonated uracil, where loss of NH_3_ dominates, and loss of HNCO is seen as a more minor product ion [59,60,61]. For protonated uracil, loss of H_2_O is also observed although we do not observe the equivalent loss of H_2_S for [2-TU·H]^+^. Notably, the major photofragment, *m*/*z* 96 (SH loss), was not seen in the HCD results (Appendix A). Further computational results on the observed fragments are presented in the Appendix A. 

To gain more insight into the photofragment production dynamics, it is useful to inspect the photofragment production spectra. These are presented in Figure 6, along with the gas-phase absorption spectrum for comparison.

Figure 6b shows the production spectrum for *m*/*z* 96 (SH loss), the most intense fragment across the entire scanned region, which displays a profile that is very similar to the gas-phase absorption spectrum (Figure 6a). The second most intense photofragment is *m*/*z* 128 (H loss), which displays the action spectrum displayed in Figure 6c. As for the *m*/*z* 96 photofragment, *m*/*z* 128 peaks strongly across the band II region (λ_max_ = 5.3 eV), although its intensity is significantly reduced across band I. Figure 6d presents the production spectrum for the *m*/*z* 70 photofragment (HNCS loss). This spectrum is very like that of *m*/*z* 96, although an additional region of production is also visible in the low-energy region between 3.8–4.2 eV. It is notable that the *m*/*z* difference between *m*/*z* 128 (Figure 6c) and *m*/*z* 70 (Figure 6d) is 58, which corresponds to the SCN unit. It may be that *m*/*z* 128 has a propensity to fragment into *m*/*z* 58 over the low-energy region. Indeed, *m*/*z* 128 is observed to fragment into lower mass channels at higher HCD energies. Finally, the spectrum for the *m*/*z* 112 photofragment is shown in Figure 6e. This is similar to the *m*/*z* 70 fragment spectrum (Figure 6d), although the relatively lower intensity of the fragment at higher energies may reflect the fact that it fragments more readily than the other fragments at higher internal energies. 

#### 2.3.3. Comparison of Photofragmentation and HCD Fragmentation of [2-TU·H]^+^

When the photofragments observed match the fragments obtained by thermal dissociation of the ground-electronic state as in HCD, the situation is described as “statistical decay”. In contrast, if dissociation occurs directly from the excited state without the involvement of a conical intersection to return the system to a near-starting point geometry, “non-statistical” decay occurs [62]. In non-statistical decay, the photofragments obtained will be notably different in their identities and relative intensities from the ground electronic states fragments obtained from HCD thermal dissociation. Our measurements on [2-TU·H]^+^ show a striking difference in the relative intensities of the photofragments compared to the HCD fragments. In the region of band I, the photofragments display relative intensities of the order *m*/*z* 70 > *m*/*z* 128 > *m*/*z* 112, while in band II region, the order changes to *m*/*z* 128 > *m*/*z* 70 > *m*/*z* 112. These relative intensities compare to the HCD relative intensities of *m*/*z* 112 > *m*/*z* 70 > *m*/*z* 128 in the HCD fragments (Appendix A). These differences in intensity, particularly given that *m*/*z* 96, a purely photochemical fragment, is the major photofragment, indicate that non-statistical decay is dominant. The observation of such non-statistical decay for the isolated, gas-phase ion, is consistent with the photophysical behaviour of 2-TU in solution [36].

## 3. Further Discussion

### 3.1. Deprotonated 2-Thiouracil

As discussed above, the gas-phase absorption spectrum of [2-TU-H]^−^ is characterised by two regions which we labelled above as band I and band II. Band I was linked to the existence of a dipole-bound excited state, in the region of the electron detachment threshold. Dipole-bound excited states can decay with the formation of either intact dipole-bound anions, or valence-bound anions (either intact or the products of dissociative electron attachment) [58,63]. For [2-TU-H]^−^, SCN^−^ is produced as a photofragment primarily in the lower-energy part of band I. This fragment has been observed in low-energy electron impact studies on 2-thiouracil [39], suggesting that the initially formed dipole-bound excited state of [2-TU-H]^−^ decays via formation of a temporary negative ion, which produces SCN^−^ as the end ionic fragment through a similar molecular process as 2-TU.

The difference in the widths of the SCN^−^ production spectrum versus the width of band I in the photodepletion spectrum is intriguing. One explanation would be that the dipole-bound excited state lies just below an electronic transition of [2-TU-H]^−^. This transition would then lie within the free-electron continuum as it is above the VDE (3.82 eV), and can therefore decay via electron detachment, leading to photodetachment rather than photofragmentation. Indeed, the TDDFT calculations predict that [2-TU-H]^−^ displays an electronic absorption at 3.8 eV, in line with this interpretation.

Band II lies fully within the electron detachment continuum and is therefore expected to correspond largely to direct electron detachment. However, some absorption is likely to be associated with the π-π* transition predicted by the TDDFT calculations at 4.73 eV (Appendix A). This excitation is clearly visible in the *m*/*z* 93 photofragment action spectrum shown in Figure 4c, indicating that the *m*/*z* 93 photofragment is produced through direct decay of the excited state accessed in this region. 

The pattern of photofragment action spectra observed here for the *m*/*z* 58 and *m*/*z* 93 photofragments is reminiscent of behaviour we observed recently in studies of iodide-nucleobase complexes [55,56,64]. These complexes similarly display a dipole-bound excited state in the vicinity of the VDE, which decays with production of the respective valence anion produced upon low-energy electron attachment to the nucleobase. At higher energies, a nucleobase-localized electronic transition occurs, which decays primarily with production of a second photofragment. Although two distinctive photofragments would be expected to be produced by these two very different excited states, we observed that both photofragments are produced in the regions of both excited states. This phenomenon appears to be unique to negative ions, and likely reflects coupling of the two excited states via the electron detachment continuum [65,66].

The dominance of electron detachment following photoexcitation of [2-TU-H]^−^ across the UV and hence, free radical production, leads to questions as to whether similar decay pathways occur in the condensed phase. Future work would be useful to directly explore this point.

### 3.2. Protonated 2-Thiouracil [2-TU·H]^+^

While the gas-phase absorption spectrum of protonated 2-TU is very similar to that of protonated uracil, its photofragmentation pathways are dramatically different. [U·H]^+^ decays with production of the statistical fragments observed in thermal decay of the ground-state system while [2-TU·H]^+^ photodecays primarily via ejection of HS·. This is true for both the band I and band II regions, corresponding to the enol-enol and enol-keto tautomers. Thus, it appears that the introduction of the S atom on moving from U to 2-TU perturbs the excited state surfaces of both 2-TU tautomers so that access to conical intersections facilitating ultrafast decay to the respective electronic ground states is prohibited. This behaviour in the protonated system appears to closely mimic that of neutral thiouracil [20]. 

Inspection of the major bright transition for the P1a tautomer of [2-TU·H]^+^ at ~5.2 eV reveals that this π → π* transition corresponds to a reduction of electron density around the sulphur atom (Appendix A), promoting C-S photochemical bond fission. The appearance of *m*/*z* 128 (H atom loss) as a photochemical fragment is also notable, as photoinduced H loss is a common decay channel for gaseous nucleobases. A similar change in electron density for the band **I** transition is also predicted by the TDDFT calculations of the P2 tautomer.

## 4. Materials and Methods

The gaseous ion absorption and photofragment spectra of [2-TU-H]^−^ and [2-TU·H]^+^ were recorded in vacuo using laser-photodissociation action spectroscopy. UV photodissociation experiments were conducted in an AmaZon quadrupole ion-trap mass spectrometer (Bruker, Billerica, MA, USA) modified for laser experiments as described previously [43,67]. UV photons were produced by an Nd:YAG (10 Hz, Surelite, Amplitude Laser Group, San Jose, CA, USA) pumped OPO (Horizon, Amplitude Laser Group, San Jose, CA, USA) laser, giving ~1 mJ across the range 390–234 nm (3.2–5.3 eV) and 214–344 nm (3.6–5.7 eV) for deprotonated and protonated 2-TU, respectively, using a 2 nm laser step size.

2-Thiouracil (99%) was purchased from Acros Organics (Loughborough, UK) and used without further purification. Methanol solutions (1 × 10^−6^ mol dm^−3^) of protonated and deprotonated 2-TU were produced by addition of a drop of trifluoroacetic acid or 2 mL of ammonium hydroxide (30%), respectively. The solutions were introduced into the mass spectrometer by electrospray ionisation (ESI) using a nebulizing gas pressure of 9 and 13 psi, an injection rate of 0.25 and 0.35 mL h^−1^, a drying gas flow rate of 8 and 3 L min^−1^, and run in positive and negative ion mode at capillary temperatures of 110 °C and 140 °C, respectively. Photofragmentation experiments were run with an ion accumulation of 100 ms with a fragmentation time of 100 ms (one laser pulse interacts with each ion packet), thereby limiting multiphoton processes. UV excited gaseous ions can fragment following excitation and produce a gas-phase absorption spectrum by photodepletion [67,68,69] for systems where fluorescence is negligible [70]. Photodepletion (PD) and photofragment production (PF) were calculated using Equations (4) and (5), respectively:(4)$$\mathrm{Photodepletion intensity}=\frac{\ln(\frac{{\mathrm{Int}}_{\mathrm{OFF}}}{{\mathrm{Int}}_{\mathrm{ON}}})}{\lambda \times P}$$
(5)$$\mathrm{Photofragmentation production}=\frac{\left(\frac{{\mathrm{Int}}_{\mathrm{FRAG}}}{{\mathrm{Int}}_{\mathrm{OFF}}}\right)}{\lambda \times P}$$
where IntON and IntOFF are the intensities of the parent ion signals with and without irradiation, respectively, and IntFrag is the ion intensity of each individual photofragment at a particular wavelength. The PD and PF intensities were taken from an average of three runs at each wavelength studied. Fragment ions with *m*/*z* < 50 fall outside the mass window of our ion trap and thus are not detectable in our mass spectrometer. 

Higher-energy collisional dissociation (HCD) experiments were performed in an Orbitrap Fusion Tribrid mass spectrometer (Thermo Fisher Scientific, Waltham, MA, USA) to acquire a wider fragmentation profile for the ground electronic states, as described previously [54,71,72]. In these experiments, the following settings were employed: spray voltage, 3600 (−4000) V; sweep gas flow rate, 1 arb.; sheath gas flow rate, 10 arb.; aux gas flow rate, 5 arb.; ion-transfer tube temperature, 275 (325) °C; vaporizer temperature, 350 °C. 

Density functional theory calculations were performed at the B3LYP/6-311++G(2d, 2p) level of theory using Gaussian 09 on a range of tautomers of [2-TU-H]^−^ and [2-TU·H]^+^ [73]. Frequency calculations were performed to ensure that the optimized structures correspond to true energy minima. Time-dependent density functional theory (TDDFT) calculations (50 states) were performed to calculate the gaseous excited state spectra, with implicit methanol solvent being used to obtain the corresponding solution-phase spectra.

## 5. Conclusions

Laser photodissociation spectroscopy of the deprotonated and protonated forms of the non-natural nucleobase, 2-TU, was performed in the gas-phase for the first time. The gas-phase absorption (photodepletion) spectra of [2-TU-H]^−^ and [2-TU·H]^+^ are highly distinctive. Whereas the gaseous absorption spectrum of [2-TU-H]^−^ displays features that can be attributed to the propensity of the negative ion to photodetach above its electron-detachment threshold, the corresponding spectrum of [2-TU·H]^+^ more closely resembles the ions’ solution-phase absorption spectrum [36]. The photodecay pathways of the protonated and deprotonated ions are also highly distinctive, with the deprotonated system producing only a small number of very low intensity fragments whereas the protonated system decays with extensive fragmentation.

Previous studies on the photophysics of thiouracil compared to uracil have found that thiolation perturbs the ability of the nucleobase to dissipate harmful UV excitation. Theoretical studies have shown that this occurs due to the initially populated bright S_2_ state decaying into the T_1_ state [29,30]. Similar photophysics appears to be present for protonated thiouracil, since the major photoproducts correspond to radical species that are indicative of dissociative triplet state decay. While the behaviour of protonated 2-TU mirrors that of neutral TU [36], our study is the first where the dissociative photoproducts have been identified. Knowledge of the identity of these photoproducts is important for assessing the suitability of thiouracil as a biochemical probe, as well as understanding its mechanistic behaviour as a photopharmaceutical.

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
