# Peer review of "Observation of Enhanced Dissociative Photochemistry in the Non-Native Nucleobase 2-Thiouracil"

_molecules, 2020, doi:10.3390/molecules25143157_

Round 1

Reviewer 1 Report

 The paper studies UV and ion-mass spectra of 2-thiouracil in the protonanted and deprotonated forms, including experiment and theory.

 The paper should be accepted after clarifications on the procedure and discussion.  In general terms is well written but there are many details missing or oversimplified that make readout difficult.  In my opinion the order of presentation mixes gas phase results with ionization and some other parts. A presentation in a more traditional way, i.e. methodology, theory and experiment, results and discussion would, in my opinion, greatly benefit the work.

  Some comments to consider:

 In 2.1, only the (large) energy difference between ground state and higher energy tautomer is the argument used to justify D1 production after electrospray in methanol, in solution (apparently using implicit solvent in gaussian) results not shown are said to be similar. I think this argument should be a bit more elaborated as there are tautomers closer in energy (cf. fig.1 and fig2 region below 4.8 eV).

 A good estimation of (vertical) dipole moment is difficult, however there are no comments on the differences or the results obtained.  Differences between D2 and D3a and why is the last one so different from D3b.  The comparison with Ref.[25] should include those results in Table 1. 

 Line 204, what is CH3I and C-I stretch (iodine?)

 A better description of the experimental procedure, with the sequential preparation of product, equipment used etc.. would help to better follow the work.  The reason for using HCD (higher-energy collisional dissociation) is not fully justified, neither the acronyms is defined until section 4.

Author Response

The reviewer comments:

In my opinion the order of presentation mixes gas phase results with ionization and some other parts. A presentation in a more traditional way, i.e. methodology, theory and experiment, results and discussion would, in my opinion, greatly benefit the work.

We would respectfully disagree with this view of the reviewer, since the order of our paper follows many others we have published previously, and is a standard format in our research field.  See for example another paper we published in Molecules: https://doi.org/10.3390/molecules23082036

Other comments:

  1. In 2.1, only the (large) energy difference between ground state and higher energy tautomer is the argument used to justify D1 production after electrospray in methanol, in solution (apparently using implicit solvent in gaussian) results not shown are said to be similar. I think this argument should be a bit more elaborated as there are tautomers closer in energy (cf. fig.1 and fig2 region below 4.8 eV).

Response: Our conclusion that only a single isomer is likely to be present is based on quite extensive work by us and other authors.  Ref [43] (referred to at line 81) of the current manuscript gives further details and further references on the topic of the appearance of protomers following ESI in protic solvents.  (The bands labelled I and II in Fig. 2 do not correspond to different isomers, as the reviewer seems to be suggesting.  Their assignment is discussed between lines 140-151.)

  1. A good estimation of (vertical) dipole moment is difficult, however there are no comments on the differences or the results obtained.  Differences between D2 and D3a and why is the last one so different from D3b.  The comparison with Ref.[25] should include those results in Table 1. 

Response:  The reviewer makes a good point about the challenges of calculating vertical dipole moments.  On reflection, since these values are not needed for our discussions in the paper, so we have deleted them from Table 1. (As spin contamination error had affected the value for D2.)

  1. Line 204, what is CH3I and C-I stretch (iodine?)

Response: All of the details about CH3I and the C-I stretch (yes, I is iodine) are given in full in reference [58]. This is perhaps quite a technical comment, so we have now placed the sentence in parentheses to show that it is not essential to the main thrust of the argument. (Lines 203-205).

  1.  A better description of the experimental procedure, with the sequential preparation of product, equipment used etc.. would help to better follow the work.  The reason for using HCD (higher-energy collisional dissociation) is not fully justified, neither the acronyms is defined until section 4.

Response: We have given quite an extended overview of our custom, experimental instrument, as is normal in physical chemistry.  Our practice in this paper is in line with our common practice as in https://doi.org/10.3390/molecules23082036, to provide one example.  We note that there is no preparation of a product in our study, as clearly stated at line 381.

The referee is correct to point out that we had not defined HCD at the point at which it is introduced in the manuscript – lines 180 and 194.  We have now defined the acronym at this point.

Reviewer 2 Report

Manuscript ID: molecules-833244

Title: Observation of Enhanced Dissociative Photochemistry in the Non-native Nucleobase 2-Thiouracil

The manuscript by Uleanya et al presents a detailed study of the dissociative photochemistry of 2-thiouracil, a nucleobase analogue in which an oxygen atom was replaced by a sulphur atom. The molecule may have some applications in pharmacology, specially in phototherapeutic treatments and therefore the results from this study may have some applications in the field.

The authors produce both the cation and the anion and study their photophysics and the fragmentation routes using LIMS and HCD. Comparison with DFT predictions allow the authors to interpret the experimental results. Discussion is rather limited, and the manuscript is mainly a description of the fragments obtained in each experiment and the possible mechanisms behind such observations. The text is well organized and well written. I am not particularly excited with the manuscript but I guess it may be of interest for the people in the field. Therefore, I recommend publication with minor revision:

  1. p5, l160. It seems to me that the peak is at 3.8 instead at 3.9. Could the authors please revise this?
  2. p7, l202. Where is the partly-solved vibronic structure of SNC-? I do not see it in Fig 4b. May be the figure requires a zoom?

Author Response

  1. p5, l160. It seems to me that the peak is at 3.8 instead at 3.9. Could the authors please revise this?

Response: The text at l160 has now been changed to 3.8

  1. p7, l202. Where is the partly-solved vibronic structure of SNC-? I do not see it in Fig 4b. May be the figure requires a zoom?

(The reviewer is referring to “partly-resolved vibronic….”

Having reviewed the presentation, we agree with the reviewer that this structure is not clear, so we have revised the text at l202 to say that the production spectrum appears to be broadened, instead of saying that it displays unresolved vibronic structure.  The text now reads:

“…production spectrum appears to be broadened on the high-energy side of the peak, possibly due to unresolved vibrational features.”

Reviewer 3 Report

The manuscript by Uleanya et al. is an impressive spectroscopic study of the photochemistry of 2-thiouracil, which is an important nucleobase analogue.

The experiments are conducted judiciously and the conclusions follow well from the returned results. The only thing that caught my eye was the presentation of equations 4 and 5 in the proof version. They have borders around them and should be revised.

I, therefore, recommend publication of this manuscript (almost as is) after this cosmetic issue has been resolved.

Author Response

  1. “The only thing that caught my eye was the presentation of equations 4 and 5 in the proof version. They have borders around them and should be revised.”

Response: This has now been corrected at lines 394 and 395